# Optimization Method for Space-Based Target Detection System Based on Background-Oriented Schlieren

**DOI:** 10.3390/s24092731

**Published:** 2024-04-25

**Authors:** Kang Li, Feng Zhou, Yun Su, Weihe Ren, Yue Zhang, Jiaquan Deng, Ruiyan Shan

**Affiliations:** 1Beijing Institute of Space Mechanics & Electricity, Beijing 100098, China; likang_0714@163.com (K.L.); suedul@163.com (Y.S.); rwh970603@163.com (W.R.); yue3723302@126.com (Y.Z.); dengjq0508@163.com (J.D.); shanry1996@163.com (R.S.); 2School of Integrated Circuits, Beijing University of Posts and Telecommunications, Beijing 100098, China

**Keywords:** background-oriented schlieren, space-based target detection, sensitivity, image signal-to-noise ratio

## Abstract

Currently, the visual detection of a target’s shock flow field through background schlieren technology is a novel detection system. However, there are very few studies on the long-distance background schlieren imaging mechanism and its application in system design in the field of target detection. This paper proposes a design optimization method for space-based BOS detection system metrics. By establishing sensitivity evaluation models and image signal-to-noise ratio evaluation models for BOS detection systems, the influence of the different flight parameters and key parameters of BOS systems (detection spectral bands and spatial resolution) on target detection efficiency is explored. Furthermore, an optimization method based on the image signal-to-noise ratio of the BOS system and the overall metrics for specific scenarios are provided. The simulation results demonstrate that under satellite background images and speckle background images, the system metrics can detect and identify the schlieren of high-speed targets, with better applicability to disordered and complex real background images. This research contributes to advancing the development of high-speed target detection technology based on BOS.

## 1. Introduction

In recent decades, with the significant improvement in the survivability of aircrafts, traditional infrared and radar detection methods are facing serious efficiency problems [1,2,3]. In fact, when an aircraft flies in the atmosphere, in addition to detecting the high temperature caused by the intense friction between the target body and the atmosphere, the interaction between the target and the atmosphere will form a high-temperature, high-pressure shock wave flow field. The use of Background-Oriented Schlieren (BOS) technology to visualize invisible flow fields is a novel detection system [4,5]. For fixed wavelength conditions, changes in gas temperature, composition, and pressure in the shock flow field will cause changes in the refractive index of the atmosphere. This leads to non-uniformity in the refractive index distribution in the shock wave region. According to the law of refraction, light will be deflected when it propagates in an environment with a non-uniform gas refractive index. Background schlieren imaging technology visualizes this non-uniform deflection phenomenon [6,7]. For example, NASA used two training jets in Armstrong in 2019 to visualize the process of aircraft turbulence interaction, verifying the feasibility of this detection technology [8].

Moreover, compared with traditional technology, background schlieren technology can significantly reduce the spatial resolution requirements of the system while maintaining the same detection performance. For example, under normal circumstances, in order to achieve target detection, it is often necessary to have a resolution that can reach the meter level or even the sub-meter level. As for the background schlieren technology, the detection scale is dozens of times larger than the target body, so the resolution can even be reduced to tens of meters, which in turn allows the system to further reduce the aperture [9,10]. Therefore, this technology has huge advantages in reducing system volume, weight, and complexity, while maintaining detection performance.

Background schlieren technology has evolved a lot. However, there are very few related studies that apply this technology to the field of target detection. The overall index design method of the background schlieren detection system is also very different from the design method of the traditional target detection system. This is specifically reflected in the following points: Firstly, their imaging principles are different: the former provides intuitive visual imaging of static scenes, while the latter images changes in scenes. Secondly, the imaging targets are different: the former aims to photograph the target body or ground scene, while the latter targets invisible flow fields for imaging. Finally, the imaging linkages are different: both remote sensing optical imaging systems and long-distance background schlieren systems perform imaging under long-distance conditions. The transmission link of the former generally considers the characteristics of target body reflection or radiation, atmospheric transmission attenuation, and optical system effects. The latter, in addition to including traditional transmission linkages, also needs to consider the light deflection characteristics of the background passing through the target flow field and the atmospheric natural flow field. Therefore, the traditional index system design method for remote sensing imaging detection systems is not applicable to background schlieren detection systems.

The main focus of this paper is to investigate an optimization method for target detection using the background shadowing detection principle. By establishing sensitivity evaluation models and image signal-to-noise ratio evaluation models for BOS detection systems, the disturbed flow field of high-speed targets is analyzed and simulated. The numerical ray tracing method is used to study the target characteristics under different scenarios and different index parameters in depth, analyze the impact of key parameters of Background-Oriented Schlieren detection systems on schlieren performance, and further study the index optimization method of space-based Background-Oriented Schlieren detection systems. This aims to provide pre-guidance and evaluation for the design of long-distance Background-Oriented Schlieren imaging systems, greatly reducing the manpower and material resource costs of system design and development.

The structure of this paper is as follows: Section 2 primarily investigates the optimization method of the BOS detection system indicators based on sensitivity evaluation models and image signal-to-noise ratio evaluation models. Section 3 mainly discusses the analysis of the impact of BOS detection system index parameters. Section 4 elaborates on the index design process, provides the system indicators for conventional space-based detection scenarios, and simulates the system imaging performance under these indicators. Additionally, a scaled-down bullet experiment was designed to verify the detection effectiveness of this technique. Lastly, conclusions are drawn in Section 5.

## 2. Sensitivity and Image Signal-to-Noise Ratio Evaluation Models for Background-Oriented Schlieren Detection Systems

### 2.1. Working Principle of BOS Detection System

Background-Oriented Schlieren (BOS) imaging utilizes the perturbation of light waves by airflow and exploits the relationship between gas refraction and flow field density gradients to perform optical non-contact flow detection. It captures imaging of non-uniform refraction phenomena in transparent media that are typically invisible to the naked eye. When a beam of light passes through a turbulent flow field, if the flow field is disturbed in some way, the light rays deviate from their original propagation direction compared to when the flow field is undisturbed. Figure 1 illustrates the wavefront distortion produced by turbulent flow fields as a plane wave passes through, where the refractive index is denoted as *n*_0_(*x,y,z*) for the incident plane wave *S*. After passing through the turbulent flow field with velocity *V_c_* and refractive index *n*(*x*,*y*,*z*), various vortices in the flow field cause deflection of the light rays. Consequently, the light ray path changes to *S′*, not only shifting in position but also altering its direction *ε* at the distorted wavefront position [11,12].

The workflow of target detection using background schlieren is shown in the Figure 2. The first step is to image the scene using a background schlieren detection camera. The camera can image the state with and without disturbance and acquire an image sequence. The image sequence contains at least background images without flow field changes and images with flow field disturbance changes. Finally, by performing optical flow calculation on the two images, the changing displacement vectors are obtained. It is possible to visualize the deflection of light in areas with disturbed gradients. In this way, the target can be detected by detecting the disturbance that is caused by the target without observing the target itself.

### 2.2. Sensitivity Evaluation Model for BOS Detection System

The Figure 3 illustrates the optical setup of the Background-Oriented Schlieren (BOS) detection system. On the far left is the representation of the natural background, while on the far right is the BOS detection camera that is used for photography. In between them lies the flow field generated by the high-speed target. *L_A_* represents the distance between the BOS camera and the flow field, *L_B_* is the distance between the BOS camera and the background, *L_D_* is the distance between the flow field and the background plate, and *ε* denotes the deflection angle of the light rays in the diagram. Under two conditions—without the flow field and with the flow field—the displacement of the light rays in the y-direction of the BOS camera’s photosensitive surface is known. The focal length of the BOS camera lens is denoted by *f*.

Based on geometric relationships, the displacement of the image points can be obtained as follows:(1)$$\Delta =ML_{D}\tan\varepsilon$$
where the magnification factor is
(2)$$M=L_{i}/L_{B}$$

Under typical circumstances, where the deflection angle ε is very small, it can be approximated that ε≈tanε. Thus, the equation can be rewritten as follows [13]:(3)$$\Delta =L_{D}\varepsilon L_{i}/L_{B}$$

In addition, the relationship between the refractive index of a fluid and its density can be simplified to the Gladstone–Dale equation [14]:(4)n−1=Kρ
where *n* is the refractive index, *K* is the Gladstone–Dale constant, and *ρ* is the density. It provides a quantitative relationship between the refractive index of a gas and its density.

From the perspective of geometrical optics, when light propagates in a non-uniform medium with a refractive index *n*, the trajectory of the light rays can be determined using Fermat’s principle (the optical path between two points is an extremum), and the deflection angle of light rays forms the refractive angle, which is equal to the integral of the refractive index gradient along the light path [15].
(5)$$\frac{d}{ds}\left(n\frac{d\vec{r}}{ds}\right)=\nabla n$$

Therefore, Background-Oriented Schlieren (BOS) detection technology actually visualizes detection imaging through the deflection disturbance that is formed by the refractive index gradient of the flow field. The deflection angle of the light rays passing through the flow field is given by
(6)$$\varepsilon =\frac{1}{n_{0}}\int \frac{\partial n}{\partial s}dz$$
where *n*_0_ is the refractive index of the outer air of the flow field, and *n* is the refractive index of the flow field. To capture clear images, the BOS camera should focus on the background. According to the relationship between the object distance and the image distance, we can obtain the following:(7)$$\frac{1}{f}=\frac{1}{L_{i}}+\frac{1}{L_{B}}$$

Combining the above equations, the relationship between the displacement offset of the image points and the refractive index of the flow field can be obtained as follows [16,17]:(8)$$\Delta =(\frac{L_{D}}{L_{D}+L_{A}-f})\frac{f}{n_{0}}\int \frac{\partial n}{\partial s}dz$$

During the actual detection process of the BOS system, the objective of the Background-Oriented Schlieren detection system is to capture the disturbance of the flow field. This is primarily achieved by detecting the magnitude of movement of the background pattern (often within the sub-pixel range). Hence, the imaging performance of the system can be characterized by the sensitivity of the BOS system [18].

The sensitivity of the BOS detection camera can be defined as the ratio of the deflection angle of light rays to the displacement of the background pattern:(9)$$S=\frac{\Delta }{\varepsilon }=ML_{D}=(\frac{L_{D}}{L_{B}-f})f$$

According to the sensitivity model formula, the sensitivity is related to the detection distance, target height, and system’s focal length, as shown in the Figure 4.

According to the above figure, for the detection camera, its sensitivity is unrelated to the deflection angle ε of the light rays. The angle ε only represents the disturbance characteristics of high-speed target schlieren. The magnitude of the image point displacement is only related to the imaging system’s focal length *f*, the distance between the flow field and the background *L_D_*, and the distance between the detection system and the background *L_B_*, independently of other factors. In other words, the sensitivity is only related to the imaging system’s focal length *f*, the distance *L_D_* between the flow field and the background, and the distance *L_B_* between the detection system and the background.

### 2.3. BOS Detection System Image Signal-to-Noise Ratio Evaluation Model

The signal-to-noise ratio generally refers to the ratio of the target signal to the noise. For Background-Oriented Schlieren (BOS) detection systems, the target signal refers to the intensity of atmospheric disturbances caused by high-speed flying targets, specifically manifested as the deviation of image pixels caused by light rays undergoing deflection after passing through the target disturbance area. Noise refers to the image pixel offset caused by camera noise and background noise, including phenomena such as photon noise, dark current, and readout circuit noise generated by the camera, as well as random fluctuations in the natural atmospheric flow field causing light ray deflection [19,20]. The signal-to-noise ratio determines the BOS camera’s ability to detect schlieren caused by high-speed targets. The specific image signal-to-noise ratio of the BOS detection system is expressed as follows:(10)$$\mathrm{ISNR}=\frac{S}{N}=\frac{\Delta }{\sqrt{{\delta_{\det\mathrm{ector}}}^{2}+{\delta_{background}}^{2}}}$$

Substituting Equations (6) and (8) into Equation (10), we get
(11)$$\mathrm{ISNR}=\frac{(\frac{L_{D}}{L_{D}+L_{A}-f})\frac{f}{n_{0}}\int \left(\nabla n\right)dz}{\sqrt{{\delta_{\det\mathrm{ector}}}^{2}+{\delta_{background}}^{2}}}\approx \frac{(\frac{L_{D}}{L_{D}+L_{A}-f})\frac{f}{n_{0}}K_{GD}{\left(\nabla \rho \right)}_{avg}w}{\sqrt{{\delta_{\det\mathrm{ector}}}^{2}+{\delta_{background}}^{2}}}$$

From the above equation, it can be seen that the signal-to-noise ratio of the BOS detection system is related to the target disturbance, detector noise disturbance, and random disturbance of the atmospheric background. The higher the target height *L_D_* is, the higher the displacement caused by the target disturbance is. However, as the target height increases, according to the schlieren characteristics of high-speed targets, the gradient of the target disturbance’s refractive index field decreases, which reduces the system’s image signal-to-noise ratio. Therefore, the target height, refractive index field gradient intensity, and system’s image signal-to-noise ratio are mutually constrained. Therefore, the selection of the actual detection system needs to be adaptively optimized according to the working scenario. In addition, the longer the focal length of the BOS detection system is, the higher the spatial resolution of the system is, resulting in a larger image signal-to-noise ratio.

## 3. Key Parameter Influence Analysis of BOS Detection System

### 3.1. Impact Analysis of Flight Speed on ISNR

In this study, high-speed flying wing-shaped targets are selected as the analysis objects. By modeling the shape and external flow field of the target, fluid simulation calculations are performed on its flow field. Wing-shaped target and flow field modeling are shown in Figure 5.

The target’s conventional cruising flight altitude is set at 10 km. The flight speed ranges from Mach 0.8 to Mach 2.2, with a speed interval of 0.2 Mach. Through simulation calculations, the variation in the flow field simulation results and distribution of characteristics of high-speed flying targets with flight speed is obtained, as shown in the Table 1.

The statistical analysis of the results shows the variation in the maximum density of the target and the maximum density gradient with flight speed, as depicted in the left figure of Figure 6. The relationship between the shockwave angle α, formed during high-speed flight, and the flight speed is illustrated in the right figure of Figure 6. It can be observed from the figures that as the flight speed increases, the shockwave shape becomes sharper, the flow field gradient becomes stronger, and the image signal-to-noise ratio of the detection system increases. If it is necessary to detect shockwave phenomena of a certain scale, the flight speed should be at least 1.0 Mach.

### 3.2. Impact Analysis of Flight Altitude on ISNR

By selecting a certain high-speed target with a conventional cruising speed of 1.6 Mach, and considering flight altitudes ranging from 0 to 20 km with intervals of 4 km, the simulation results and characteristic distributions of the flow field around the high-speed flying target at varying flight altitudes are obtained through simulation calculations, as illustrated in the Table 2.

Figure 7 shows the results of statistical analysis. It is observed that both the maximum density and the maximum density gradient of the target gradually decrease with the increase in flight altitude. According to the ISNR formula, it can be inferred that the image signal-to-noise ratio (ISNR) of the BOS system is proportional to the target density gradient. Therefore, for the BOS detection system, as the flight altitude increases, the disturbance caused by the target decreases, leading to a lower image detection signal-to-noise ratio. Moreover, the trend of this decrease is not linear but tends to flatten out progressively.

### 3.3. Analysis of Impact of Detection Spectral Bands on ISNR

Due to the differing principles between Background-Oriented Schlieren (BOS) detection systems and conventional optical detection cameras, the selection of spectral bands does not rely on the contrast between the target radiation features and background radiation peaks. Instead, it considers the deflection capability of gradient flow fields in the chosen spectral band. A greater deflection of light rays under the same intensity of atmospheric turbulence is more favorable for detection.

Substituting Equation (4) into Equation (11), we obtain the following:(12)$$\mathrm{ISNR}=\frac{2.2244\times 10^{-4}(\frac{L_{D}}{L_{D}+L_{A}-f})\frac{f}{n_{0}}\left(1+\frac{4.51\times 10^{-3}}{\lambda^{2}}\right){\left(\nabla \rho \right)}_{avg}w}{\sqrt{{\delta_{\det\mathrm{ector}}}^{2}+{\delta_{background}}^{2}}}$$

Normalizing the equation yields, we find that
(13)$$\mathrm{ISNR}\propto 1+\frac{4.51\times 10^{-3}}{\lambda^{2}}$$

Therefore, the influence of the detection spectral bands on the ISNR is illustrated in the Figure 8.

From the graph, it can be observed that the longer the spectral band is, the smaller the deflection caused by the disturbance of the target flow field is, resulting in a lower image signal-to-noise ratio (ISNR) for the BOS detection system. Therefore, choosing visible spectral bands is more advantageous for the system’s detection. However, the actual impact of the detection spectral bands is relatively small, with only about a 3% difference between visible and long-wave bands. However, the camera’s frame rate that is used in background schlieren imaging technology is generally higher. The current high-frame-rate detectors are very immature in the infrared spectrum. There are currently many high-frame-rate detectors in the visible spectrum. While maintaining high-frame-rate performance, its noise can reach a low level. Therefore, background schlieren cameras generally select the visible spectrum band.

Additionally, since the actual imaging object of the Background-Oriented Schlieren (BOS) detection system is the ground scene, another limiting factor for spectral band selection is the contrast characteristics of different spectral bands in ground reflectance. Due to the presence of partial angular scenes during satellite detection, this paper analyzes the contrast between the ground reflection spectral radiance and the atmospheric scattering spectral radiance under different detection angles, as shown in the Figure 9 [21].

The Background-Oriented Schlieren (BOS) detection system requires ground scene images to have high contrast. From the graph, it can be observed that in the spectral range of 0.63 μm to 0.76 μm, clearer ground patterns are more advantageous for enhancing the detection of target schlieren.

### 3.4. Spatial Resolution Impact Analysis on ISNR

The required spatial resolution of the background shadow detection system is related to the detection capability of the light ray deflection and the size of the detectable disturbance area. For the BOS detection system, detection relies on identifying disturbances in the sub-pixel resolution formed by the target. A higher spatial resolution leads to increased sensitivity of the system and better detection of target disturbances.

From the perspective of deflection angle detection, the intensity of the detectable atmospheric turbulence by the target is related to the turbulence detection sensitivity and the ground pixel resolution. The formula for calculating the target deflection intensity is
(14)GSD=LD⋅tan(ε)1N≈LD⋅ε1N
where *GSD* represents the ground pixel resolution, and 1/*N* represents the detection capability of the BOS detection system algorithm, typically taken as 0.3 based on conventional detection algorithm capabilities. Substituting Equation (6) yields the following:(15)GSD=LD⋅tan(ε)1N≈LD⋅ε1N=LDKGD(∇ρ)avgw1N⋅n0

Based on the conclusions of the influencing factor analysis in Section 3.2, the relationship between the spatial resolution of the BOS detection system and the target flight height is shown in Figure 10.

Considering the size of the detectable disturbance area, when the BOS detection system detects a winged target, simulations are conducted with a flight height of 10 km and an analytical speed of 1.6 Ma. The disturbance area generated by the winged target is shown in the Figure 11. A-A′ and B-B′ show the interference direction and longitudinal slices. From the figure, it can be observed that the disturbance area generated by the winged target contains multiple linear features. Each linear feature disturbance area gradually decreases linearly in the disturbance direction, reaching approximately one-fourth of the original intensity at a distance of about 20 times the target’s body size.

## 4. Design and Performance Simulation of Space-Based BOS Detection System

Designing the detection process of high-speed targets using the background shadow imaging system is illustrated in Figure 12. Generally, the overall performance indicators of the system need to be determined based on the detection task requirements. Then, the performance indicators are decomposed, and the spectral range and spatial resolution of the detection process are designed according to the specific scenario. Since the spatial resolution directly determines the sensitivity and accuracy of the final detected target shadow, it is a key indicator that requires special attention. Subsequently, based on the spatial resolution and device capabilities, other indicators such as the aperture and detector are determined. Once the overall performance indicators are designed, detailed design indicators and schemes can be developed for each subsystem of the BOS detection camera.

Figure 12 depicts the design process of the BOS system indicators.

In this study, considering general mission requirements, a detection probability of 90%, a false alarm rate of 10^−4^, and an operational distance of 400 km are proposed as the system performance indicators. According to signal detection theory [22,23], the relationship between the detection rate, false alarm rate, and SNR under binary detection is represented by the following equation, where *Q* is the Macsum Q function [24], *P_d_* is the detection rate, *P_fa_* is the false alarm rate, and SNR is the signal-to-noise ratio:(16)$$P_{d}=Q(\sqrt{2*SNR},\sqrt{-2*\ln\left(P_{fa}\right)})$$

The relationship between the three is illustrated in Figure 13.

Based on this curve, to meet the requirements for the detection rate and false alarm rate, the signal-to-noise ratio needs to be greater than a certain threshold. Utilizing the relationship between the target SNR and detection rate at a specific false alarm rate, under the 90%|10^−4^ detection rate|false alarm rate requirement, the signal-to-noise ratio should be at least 12 dB, that is, SNR ≥ 4. According to the analysis in Section 3.4, a spatial resolution of 10 m is more appropriate for the detection system. Subsequently, only using conventional optical camera indicator decomposition methods, the indicators of the BOS detection system can be obtained as shown in Table 3.

Using the full-chain simulation scene generation method described in the literature [25,26], the system’s performance was simulated based on the detected system metrics, obtained from the analysis above. Performance simulation was performed by using actual on-orbit satellite coastline images, city images, and black-and-white speckle images as backgrounds. This method primarily involved optical tracing of each source point ray within the wing-shaped refractive index field. Subsequently, the traced ray vectors were obtained using tracing algorithms. Weighted distribution of the attached brightness information was then performed. Finally, we could obtain a new image containing the overlaid system performance parameter information derived from the tracing process.

The results are shown in Table 4. From the table, it can be observed that both satellite background images and speckle background images can be used to detect the streaks of high-speed targets. Moreover, they exhibit better detection performance for disordered and complex background images. Hence, real and complex environments demonstrate better applicability for the BOS detection system.

In order to describe the applicability of the designed system indicators to different contexts, this article uses the correlation index C of the background image to describe the difference in the background. This indicator is used to describe the similarity of background patterns. The correlation degree C of the image is measured by the grayscale correlation of local pixels, and the definition formula is as shown in the following formula:(17)$$C=\frac{\sum_{i=1}^{N}\left(x_{i}-\frac{1}{N}\Sigma_{i=1}^{N}x_{i})(y_{i}-\frac{1}{N}\Sigma_{i=1}^{N}y_{i}\right)}{\sqrt{\sum_{i=1}^{N}{\left(x_{i}-\frac{1}{N}\sum_{i=1}^{N}x_{i}\right)}^{2}\times \Sigma_{i=1}^{N}{\left(y_{i}-\frac{1}{N}\Sigma_{i=1}^{N}y_{i}\right)}^{2}}}$$

In the equation, *N* represents the number of pixels in the grayscale image, where *x_i_* and *y_i_* are the pixel values of adjacent pixels.

We calculated the correlation between the urban background, coastline background, and speckle background. Afterwards, histogram statistical calculations were performed on the optical flow detection results under different background images. We then used histograms to characterize the precision of the detection results. The statistical results are shown in Table 5.

It can be concluded from the formula of correlation that the higher the correlation C of the image is, the smaller the difference between adjacent pixels in the image is. Among the above three backgrounds, the coastline image has the highest correlation, followed by the city image, and the speckle image has the lowest correlation. Judging from the optical flow detection results, the speckle image detection results have the largest number of detected pixels and the richest details, followed by the city images and, lastly, the coastline images. It can be concluded that for disordered and complex background images, the smaller the correlation is, the better the detection effect is. Therefore, when performing target detection, the BOS detection system designed in this article has better applicability for real complex environments.

In order to verify the detection efficiency of the 10 m spatial resolution designed in this article under the condition of 400 km detection distance, a scaling test was designed for this scenario. The test uses bullets as detection targets and conducts equivalent scale tests. Due to the limitations of the test site, the detection distance can only simulate the detection conditions of 5 m. Therefore, for a detection distance of 5 m, the spatial resolution is scaled proportionally. It is concluded that the spatial resolution should be 0.25 mm. The test site and test results are shown in Figure 14.

We performed optical flow detection on the above test images. The detection results are shown in Figure 15. It can be found from the figure that the equivalent scaling test can detect the turbulence of the bullet target in the image, which equivalently verifies the detectability of the system.

## 5. Conclusions

To address the detection challenges of high-speed airborne targets, this paper proposes a design method for a space-based Background-Oriented Schlieren (BOS) detection system. By establishing sensitivity evaluation models and image signal-to-noise ratio (ISNR) evaluation models that are specific to the BOS detection system, the study simulates and analyzes the effects of the flight speed and altitude of conventional wing-shaped high-speed flying targets on the target schlieren characteristics. It investigates the impact of different flight parameters and key parameters of the BOS system (such as the detection spectral bands and spatial resolution) on target detection performance. Additionally, it proposes a process and method for optimizing the BOS detection system parameters and provides overall system indicators based on real-space scenarios. Through ray-tracing digital simulation, this study verifies that under satellite background images and speckle background images, the system indicators can achieve the detection and identification of high-speed targets’ schlieren, with better applicability to disordered and complex real background images.

## Figures and Tables

**Figure 1 sensors-24-02731-f001:**
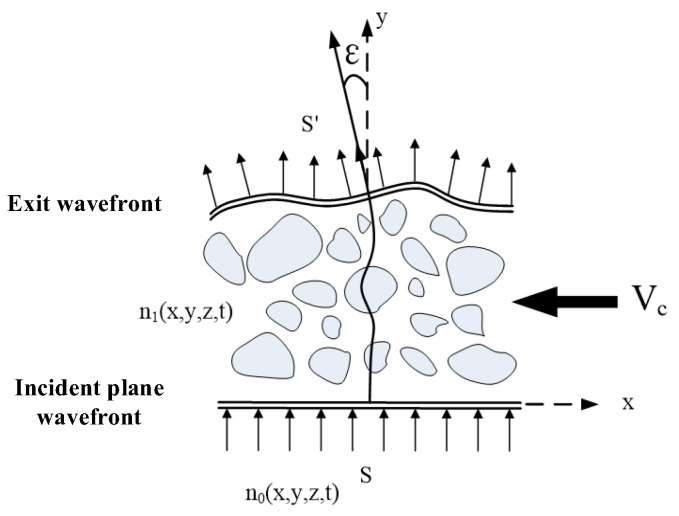
Schematic diagram of wavefront distortion generated by a turbulent flow field acting on a plane wave.

**Figure 2 sensors-24-02731-f002:**
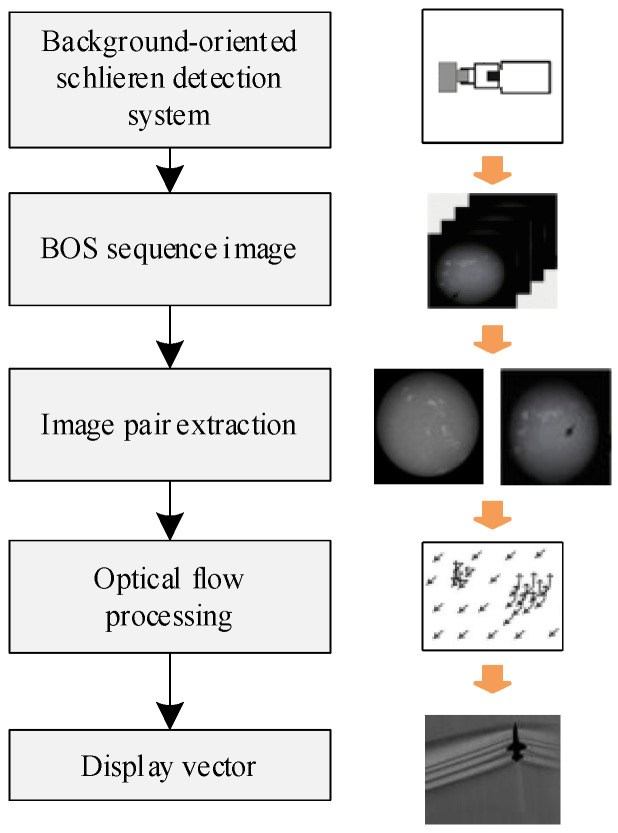
Workflow diagram of Background-Oriented Schlieren detection system.

**Figure 3 sensors-24-02731-f003:**
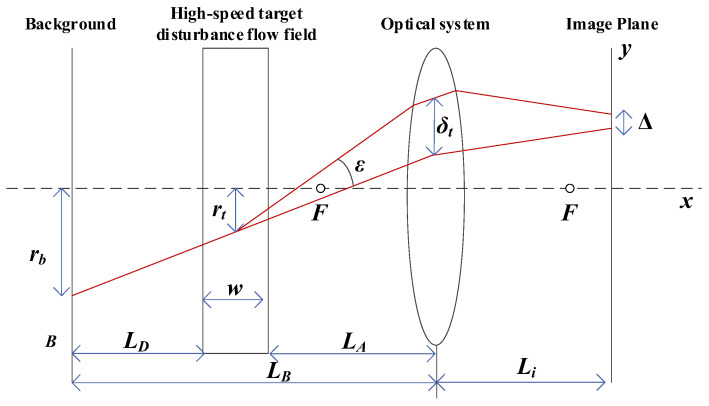
Optical configuration of BOS detection system.

**Figure 4 sensors-24-02731-f004:**
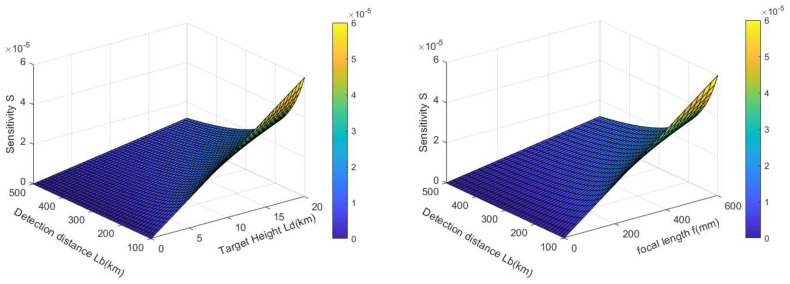
The relationship between detection distance, target height, camera focal length, and sensitivity.

**Figure 5 sensors-24-02731-f005:**
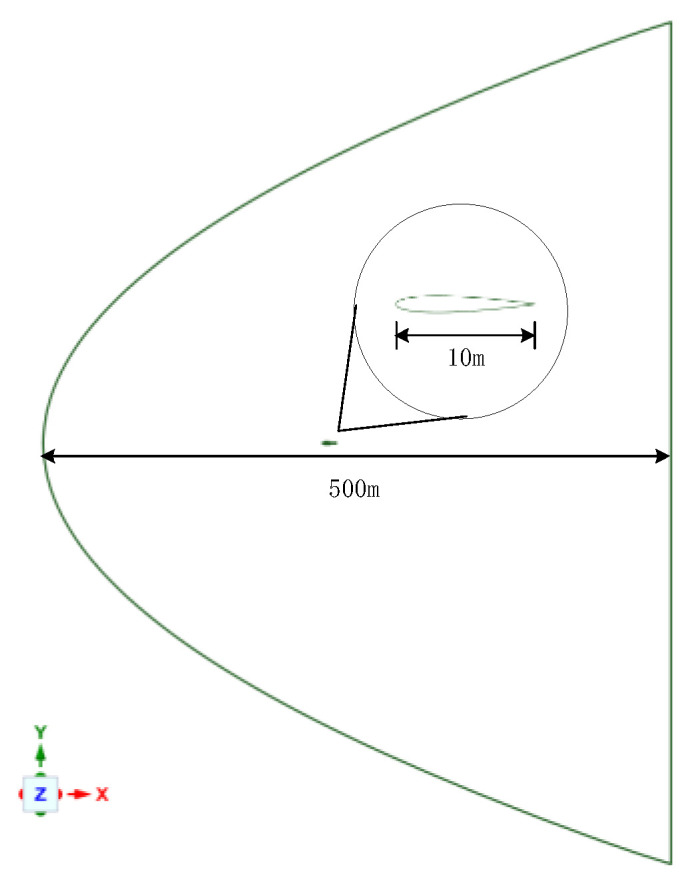
Wing-shaped target and flow field modeling.

**Figure 6 sensors-24-02731-f006:**
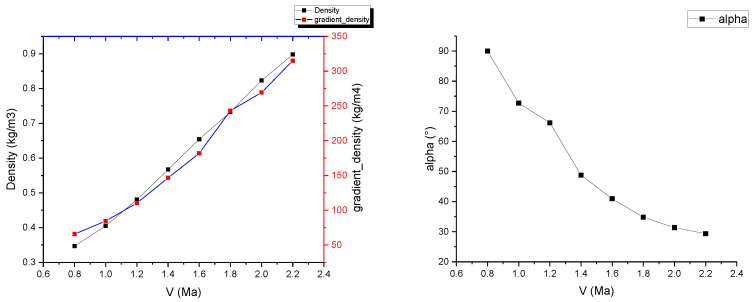
The relationship between the maximum density of the target, the maximum density gradient, and the shockwave angle and flight speed.

**Figure 7 sensors-24-02731-f007:**
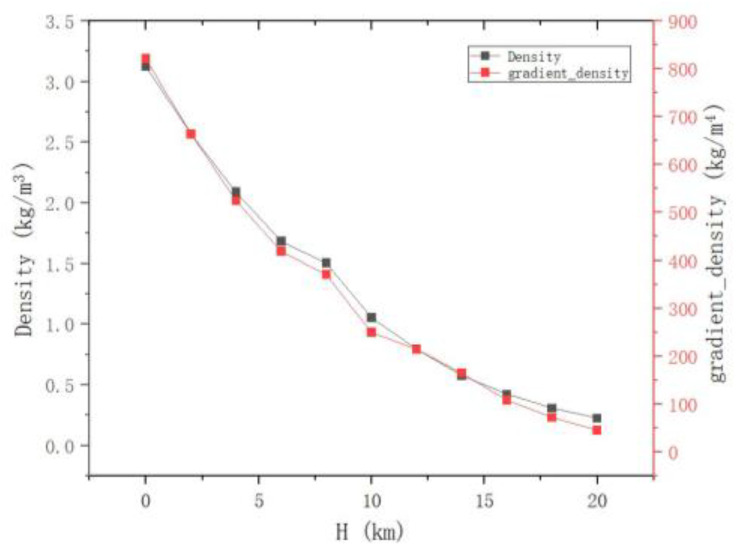
The relationship between density, density gradient, and flight altitude.

**Figure 8 sensors-24-02731-f008:**
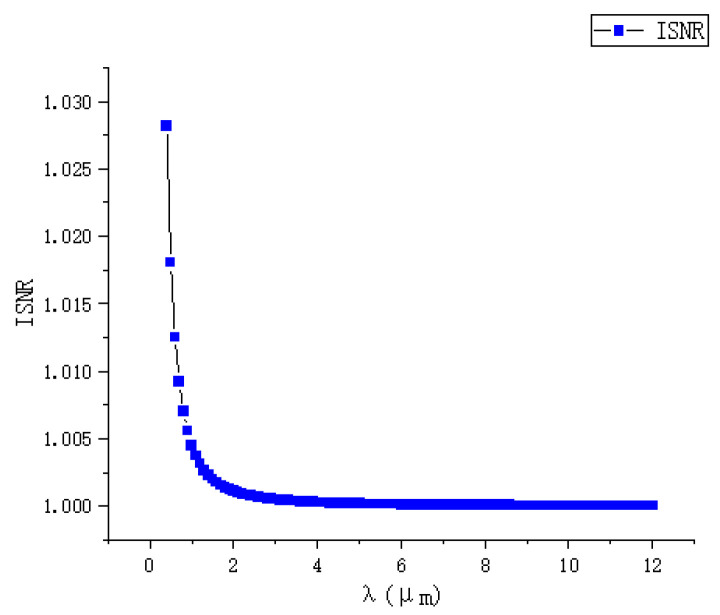
The relationship between the detection spectral bands and the normalized image signal-to-noise ratio (ISNR).

**Figure 9 sensors-24-02731-f009:**
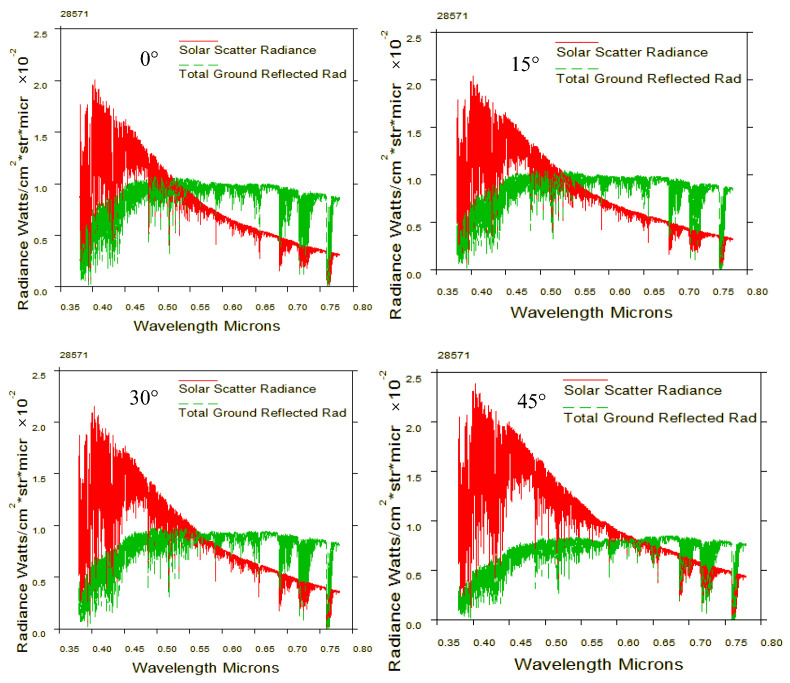
Comparison of ground reflectance and atmospheric scattering radiance at downward viewing angles of 0°, 15°, 30°, and 45°.

**Figure 10 sensors-24-02731-f010:**
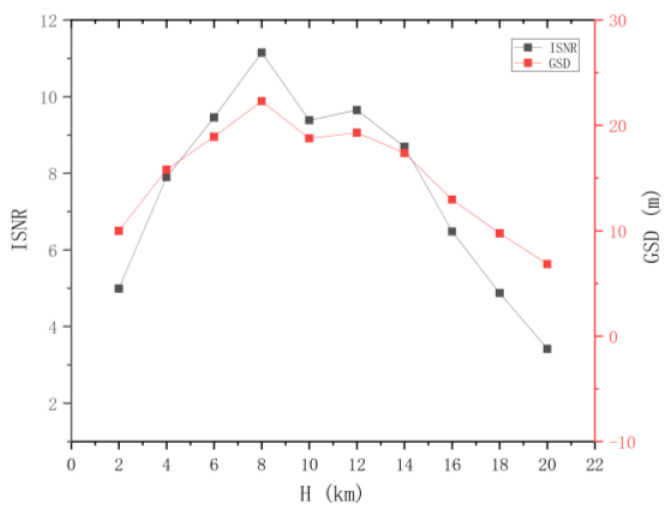
The impact of flight altitude on image signal-to-noise ratio (ISNR) and ground resolution.

**Figure 11 sensors-24-02731-f011:**
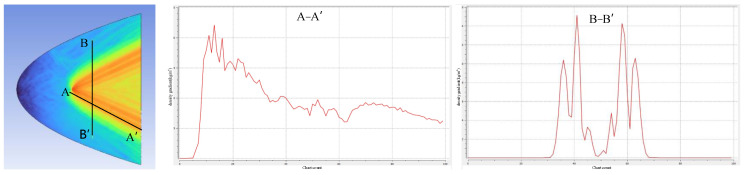
Analysis of winged target disturbance’s intensity and range.

**Figure 12 sensors-24-02731-f012:**
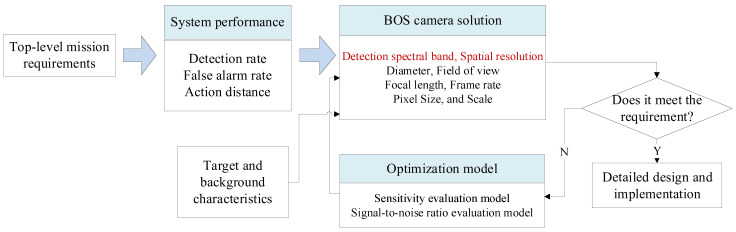
Design process of BOS system indicator framework.

**Figure 13 sensors-24-02731-f013:**
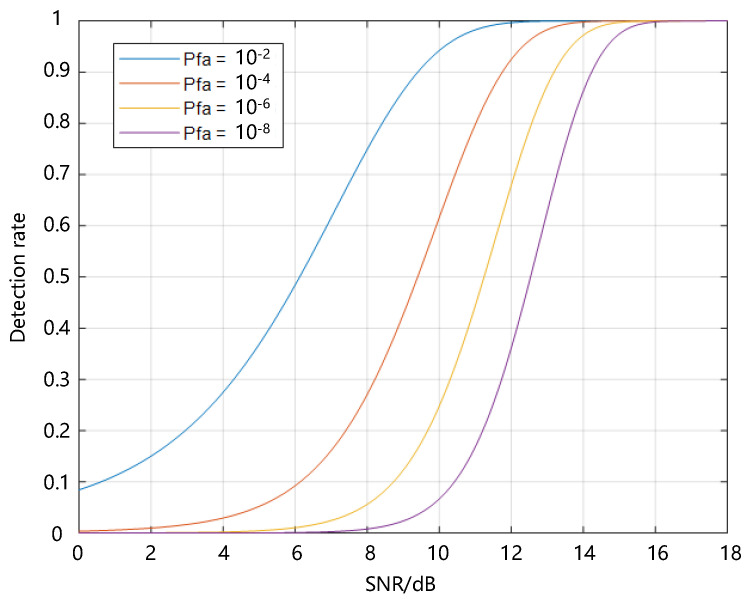
The relationship between detection rate, false alarm rate, and SNR.

**Figure 14 sensors-24-02731-f014:**
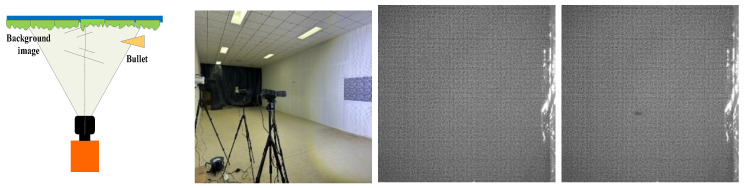
Experimental procedure and test images.

**Figure 15 sensors-24-02731-f015:**
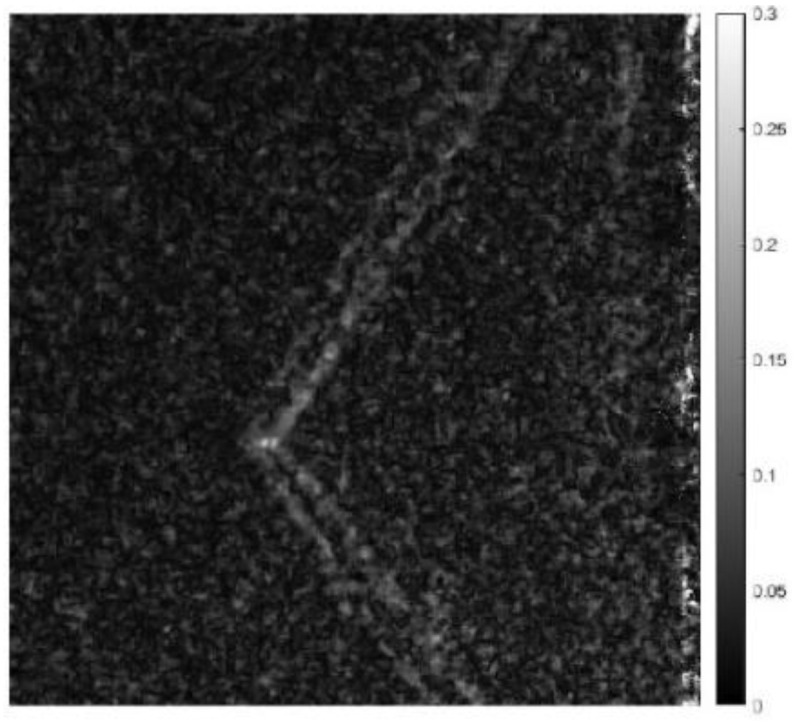
Bullet optical flow detection results.

**Table 1 sensors-24-02731-t001:** Results of flow field calculations at different speeds.

Velocity	0.8 Ma	1.0 Ma	1.2 Ma	1.4 Ma	1.6 Ma	1.8 Ma	2.0 Ma	2.2 Ma
density contour (kg/m^3^)	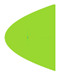	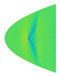	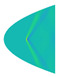	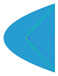	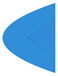	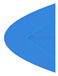	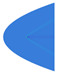	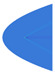
gradient contour (kg/m^4^)	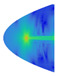	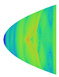	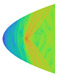	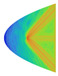	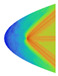	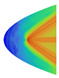	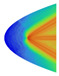	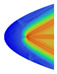

**Table 2 sensors-24-02731-t002:** Results of flow field calculations at different heights.

Height (km)	0	4	8	12	16	20
density contour (kg/m^3^)	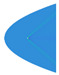	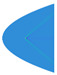	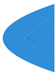	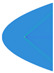	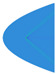	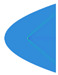
gradient contour (kg/m^4^)	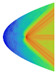	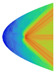	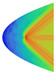	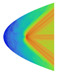	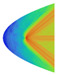	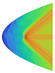

**Table 3 sensors-24-02731-t003:** BOS detection system metrics.

Metrics	Values
Orbit height (km)	400
Detected wavelength (μm)	0.63~0.76
Spatial resolution (m)	10
Diameters of pupil (mm)	50
Focal length (m)	400
Pixel size (µm)	10
Array size	1024 × 1024

**Table 4 sensors-24-02731-t004:** Performance simulation results.

Background	Original Image	Trace Generation Image	Optical Flow Detection Image
Satellite coastline image	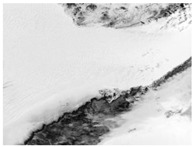	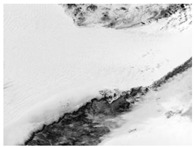	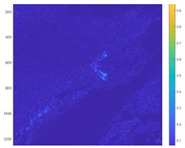
Satellite city image	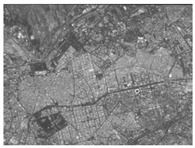	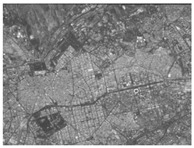	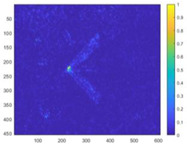
Black-and-white speckle image	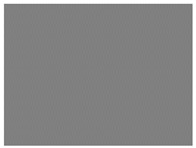	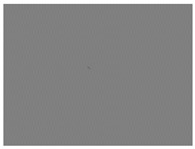	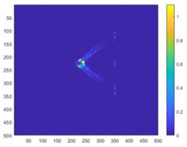

**Table 5 sensors-24-02731-t005:** Different background correlation and detection result histograms.

Background	Original Image	Image Correlation	Optical Flow Detection Image Histogram
Satellite coastline image	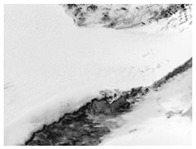	0.93	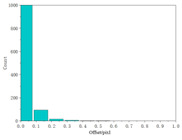
Satellite city image	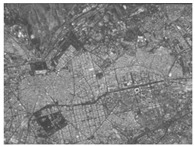	0.74	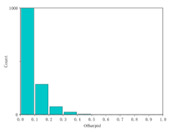
Black-and-white speckle image	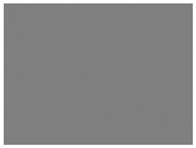	−1	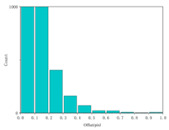

## Data Availability

The data that support the findings of this study are available on request from the corresponding author upon reasonable request.

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
