# Peer review of "Optimization Method for Space-Based Target Detection System Based on Background-Oriented Schlieren"

_sensors, 2024, doi:10.3390/s24092731_

Round 1

Reviewer 1 Report

Comments and Suggestions for Authors

The manuscript entitled "Optimization Method for Space-based High-Speed Target Detection System Based on Background Oriented Schlieren" proposes a design optimization method for a space-based BOS detection system. By establishing sensitivity evaluation models and image signal-to-noise ratio evaluation models, the manuscript explores the influence of different flight parameters and key parameters of BOS systems on target detection efficiency. I would like to pose the following queries and suggestions for the authors:

1. The manuscript presents a sensitivity evaluation model for the BOS detection system. Could the authors elaborate on how this model compares with existing sensitivity evaluation approaches in the literature? Additionally, is there a specific reason for the chosen sensitivity evaluation parameters, and how do they impact the overall detection efficiency?

2. The introduction provides a broad overview of the challenges faced by traditional detection methods. However, it would be beneficial to include a more detailed discussion on how the proposed BOS detection system addresses these challenges, particularly in the context of space-based applications.

3. The paper utilizes numerical ray tracing methods for simulation. Can the authors provide more details on the validation of these simulation models? How do the simulation results correlate with real-world experimental data, if available?

4. The manuscript mentions the system's applicability to disordered and complex real background images. Could the authors provide examples or case studies where the system has been successfully applied in such scenarios? Furthermore, how does the system's performance vary with different types of background images?

5. The selection of detection spectral bands is discussed in relation to the deflection capability of gradient flow fields. Are there any trade-offs or limitations associated with choosing visible spectral bands over longer-wave bands, considering factors like atmospheric absorption and scattering?

6. The impact of spatial resolution on the BOS detection system's sensitivity is analyzed. Could the authors comment on the optimal spatial resolution for different detection scenarios and how it affects the system's overall performance?

8. In the conclusion, the authors highlight the potential of the proposed BOS detection system. Are there any specific areas or applications where the authors envision significant advancements or improvements in future research?

Author Response

Thank you very much for your valuable comments on this article. The following are my replies and modifications to the article:

Comments 1: The manuscript presents a sensitivity evaluation model for the BOS detection system. Could the authors elaborate on how this model compares with existing sensitivity evaluation approaches in the literature? Additionally, is there a specific reason for the chosen sensitivity evaluation parameters, and how do they impact the overall detection efficiency?

Response 1: This article mainly gives the design method of the indicator system of the BOS detection system. Since there is currently no relevant research that applies the BOS principle to the field of target detection, the innovation of this method is to study an index system design method based on target detection efficiency.

For sensitivity assessment, this article starts from the principle of BOS detection and formulates the process of light transmission in the link. These formulas are a collection of multiple theories, including the theory of the relationship between atmospheric density and refractive index, geometric ray tracing theory of inhomogeneous media, paraxial optics theory, etc.

The reason for selecting the sensitivity evaluation parameters is mainly based on the key parameters in the derived sensitivity formula. For example, the sensitivity formula includes detection distance, target height, system focal length, flow field gradient intensity, etc. The flow field intensity is closely related to the target flight speed, altitude, shape, etc. Therefore, this article mainly analyzes the influence of these key parameters.

This change of manuscript can be found in Introduction section.

Comments 2: The introduction provides a broad overview of the challenges faced by traditional detection methods. However, it would be beneficial to include a more detailed discussion on how the proposed BOS detection system addresses these challenges, particularly in the context of space-based applications.

Response 2: Compared with traditional technology, background schlieren technology can significantly reduce the spatial resolution requirements of the system while main-taining the same detection performance. For example, under normal circumstances, in order to achieve target detection, it is often necessary to have a resolution that can reach the meter level or even the sub-meter level. As for the background schlieren technology, the detection scale is dozens of times larger than the target body, so the resolution can even be reduced to tens of meters, which in turn allows the system to further reduce the aperture. Therefore, this technology has huge advantages in reducing system volume, weight and complexity while maintaining detection performance.

This change of manuscript can be found – page 1, paragraph 2.

Comments 3: The paper utilizes numerical ray tracing methods for simulation. Can the authors provide more details on the validation of these simulation models? How do the simulation results correlate with real-world experimental data, if available?

Response 3: The ray tracing method for variable refractive index media has become very mature. Generally, the Runge-Kutta method is used for ray tracing. This type of numerical method has been verified in many articles, including descriptions in another article of mine and references [26] and [27], so this article does not explain the details.

Comments 4: The manuscript mentions the system's applicability to disordered and complex real background images. Could the authors provide examples or case studies where the system has been successfully applied in such scenarios? Furthermore, how does the system's performance vary with different types of background images?

Response 4: The article adds simulation results of satellite coastline background images. In addition, a description formula for image complexity is added, and image correlation is used to quantitatively describe complexity. Finally, histogram statistics are used to quantitatively describe the detection results more intuitively. To illustrate the BOS detection system has better applicability for real complex environments.

This change of manuscript can be found – page 12, paragraph 3, line 365.

Comments 5: The introduction provides a broad overview of the challenges faced by traditional detection methods. However, it would be beneficial to include a more detailed discussion on how the proposed BOS detection system addresses these challenges, particularly in the context of space-based applications.

Response 5: The camera frame rate used in background schlieren imaging technology is generally higher. The current high frame rate detectors are very immature in the infrared spectrum. There are currently many high frame rate detectors in the visible spectrum. While maintaining high frame rate performance, its noise can reach a low level. Therefore, background schlieren cameras generally select the visible spectrum band.

This change of manuscript can be found – page 9, paragraph 1, line 269.

Comments 6: The introduction provides a broad overview of the challenges faced by traditional detection methods. However, it would be beneficial to include a more detailed discussion on how the proposed BOS detection system addresses these challenges, particularly in the context of space-based applications.

Response 6: The article adds simulation results for different background images at the end. And provide a more intuitive quantitative description of the detection results. In addition, a scaled-down test more convincingly proved the effectiveness of the BOS system.

This change of manuscript can be found – page 13, line 389.

Comments 7: The introduction provides a broad overview of the challenges faced by traditional detection methods. However, it would be beneficial to include a more detailed discussion on how the proposed BOS detection system addresses these challenges, particularly in the context of space-based applications.

Response 7: The BOS detection system studied in this article is mainly intended to be used in the field of target detection. For example, the detection and discovery of targets such as aircraft and hypersonic vehicles.

Reviewer 2 Report

Comments and Suggestions for Authors

1. Please list your contribution in the introduction section of the manuscript. It seems that the manuscript is consisted of several numerical results. There lacks theoretical foundations.

2. It is necessary to summarize the flow of the BOS in a table for easy to see clearly.

3. Some numerical comparisons with existing methods should be provided. 

Comments on the Quality of English Language

The language should be polished.

Author Response

Thank you very much for your valuable comments on this article. The following are my replies and modifications to the article:

Comments 1: Please list your contribution in the introduction section of the manuscript. It seems that the manuscript is consisted of several numerical results. There lacks theoretical foundations.

Response 1: The article has reorganized and rewritten the introduction part. The research background of the article has been re-simplified. It points out the huge advantages of using BOS technology in the field of target detection. By explaining the current problems faced by applying BOS in the field of detection, it focuses on describing the innovative content of this article, which is to propose a novel BOS detection system design method that can be used for long-distance target detection. All the content of the article is to serve the final indicator design system. This design system is currently at a stage that no one has studied yet. The theoretical part of the article lies in deriving the influencing parameter formulas and influencing rules of the system when BOS is used for target detection. At the end of the article, the verification part is rewritten and simulation results and test results are added.

This change of manuscript can be found in Introduction section.

Comments 2: It is necessary to summarize the flow of the BOS in a table for easy to see clearly.

Response 2: The article adds a section on the working principle of the BOS detection system. In this section, the workflow when BOS technology is applied to target detection is explained. The article draws a more intuitive flow chart to illustrate this process.

This change of manuscript can be found – page 3, paragraph 2, line 104.

Comments 3: Some numerical comparisons with existing methods should be provided.

Response 3: In the research process of this article, CFD simulation method, variable refractive index medium ray tracing method, etc. were mainly used. The CFD simulation method is researched using the mature Fluent software. The ray tracing method for variable refractive index media has become very mature. Generally, the Runge-Kutta method is used for ray tracing. This type of numerical method has been verified in many articles, including descriptions in another article of mine and references [26] and [27], so this article does not explain the details.

In addition, in order to increase the persuasiveness of the article, in the penultimate section, The article adds simulation results of satellite coastline background images. In addition, a description formula for image complexity is added, and image correlation is used to quantitatively describe complexity. Finally, histogram statistics are used to quantitatively describe the detection results more intuitively. To illustrate the BOS detection system has better applicability for real complex environments. At last, a scaled-down test more convincingly proved the effectiveness of the BOS system.

Round 2

Reviewer 1 Report

Comments and Suggestions for Authors

The authors have incorporated the suggestions in the revised version.

Reviewer 2 Report

Comments and Suggestions for Authors

The authors have addressed all the comments.

Comments on the Quality of English Language

The language can be improved.